# MAEL Augments Cancer Stemness Properties and Resistance to Sorafenib in Hepatocellular Carcinoma through the PTGS2/AKT/STAT3 Axis

**DOI:** 10.3390/cancers14122880

**Published:** 2022-06-10

**Authors:** Chaoran Shi, Dora Lai-Wan Kwong, Xue Li, Xia Wang, Xiaona Fang, Liangzhan Sun, Ying Tang, Xin-Yuan Guan, Shan-Shan Li

**Affiliations:** 1Department of Clinical Oncology, Li Ka Shing Faculty of Medicine, The University of Hong Kong, Hong Kong SAR, China; u3005126@connect.hku.hk (C.S.); dlwkwong@hku.hk (D.L.-W.K.); fangxn@connect.hku.hk (X.F.); u3005197@connect.hku.hk (L.S.); u3005202@hku.hk (Y.T.); 2Department of Surgery, Li Ka Shing Faculty of Medicine, The University of Hong Kong, Hong Kong SAR, China; daisyxue@hku.hk; 3Department of Pathology, Li Ka Shing Faculty of Medicine, The University of Hong Kong, Hong Kong SAR, China; wxia0951@hku.hk; 4State Key Laboratory of Liver Research, Li Ka Shing Faculty of Medicine, The University of Hong Kong, Hong Kong SAR, China; 5State Key Laboratory of Oncology in Southern China, Sun Yat-Sen University Cancer Center, Guangzhou 510060, China; 6Department of Clinical Oncology, The University of Hong Kong-Shenzhen Hospital, Shenzhen 518058, China

**Keywords:** Maelstrom, MAEL, PTGS2, HCC, drug resistance, cancer stemness

## Abstract

**Simple Summary:**

Hepatocellular cancer (HCC) is the most common and lethal subtype of liver cancer without effective therapeutics. Understanding and targeting cancer stem cells (CSCs), a stem-cell-like subpopulation, which are emerging as effective ways to decipher tumor biology and develop therapies, may help to revolutionize cancer management. Cancer/testis antigen Maelstrom (MAEL) has been implicated in the regulation of CSC phenotypes, while the role of CSCs remains unclear. We demonstrated that MAEL positively regulates cancer stem-cell-like properties in HCC, and MAEL silencing provokes tumor cells’ sensitivity to sorafenib. We further discovered that the MAEL-dependent stemness was operated via PGST2/IL8/AKT/STAT3 signaling. Collectively, our study suggests the MAEL/PGST2 axis as a potential therapeutic target against CSC and sorafenib resistance in HCC.

**Abstract:**

Cancer stem cells (CSCs) are responsible for tumorigenesis, therapeutic resistance, and metastasis in hepatocellular cancer (HCC). Cancer/testis antigen Maelstrom (MAEL) is implicated in the formation of CSC phenotypes, while the exact role and underlying mechanism remain unclear. Here, we found the upregulation of MAEL in HCC, with its expression negatively correlated with survival outcome. Functionally, MAEL promoted tumor cell aggressiveness, tumor stem-like potentials, and resistance to sorafenib in HCC cell lines. Transcriptional profiling indicated the dysregulation of stemness in MAEL knockout cells and identified PTGS2 as a critical downstream target transactivated by MAEL. The suppression effect of MAEL knockout in tumor aggressiveness was rescued in PTGS2 overexpression HCC cells. A molecular mechanism study revealed that the upregulation of PTGS2 by MAEL subsequently resulted in IL-8 secretion and the activation of AKT/NF-κB/STAT3 signaling. Collectively, our work identifies MAEL as an important stemness regulation gene in HCC. Targeting MAEL or its downstream molecules may provide a novel possibility for the elimination of CSC to enhance therapeutic efficacy for HCC patients in the future.

## 1. Introduction

Hepatocellular carcinoma (HCC), accounting for nearly 80% of cases of liver cancer, is the second-most frequent cause of cancer death worldwide [1]. According to the latest Hong Kong government report, liver cancer is the fourth-most frequent cancer and the third-most common cause of cancer death [2]. Sorafenib is the first FDA-approved systemic mono-therapeutic drug for unresectable HCC [3]. In recent years, novel treatments, including tyrosine kinase inhibitors (lenvatinib, regorafenib, and cabozantinib), antiangiogenic monoclonal antibodies (bevacizumab and ramucirumab), and immunotherapies (nivolumab, pembrolizumab, and atezolizumab), have become available for HCC therapy [4,5,6,7,8,9,10]. However, their response and survival rates are still limited [11,12]. A small population of cancer cells with distinct stem cell properties, including self-renewal, differentiation, and tumorigenesis, has been identified and characterized as CSCs in most tumors [13]. Evidence has supported that HCC potentially arises from CSCs, which contribute to the formation of resistance to therapies [14,15]. Thus, a better understanding of the mechanisms driving cancer stemness is urgently needed to overcome CSCs in HCC treatment.

In HCC, CSCs have been identified by multiple well-known surface markers of stemness, such as CD133 [16], CD44 [17], CD24 [18], CD13 [19], and epithelial cell adhesion molecules (EpCAMs) [20]. Several vital regulators are responsible for the development of CSCs and the maintenance of stemness properties in HCC, including Oct4, sox2, Nanog, and NOTCH [21]. In addition, Wnt/β-catenin, IL-6/STAT3, NF-κB, and Hippo signaling pathways are involved in regulating liver cancer stemness [22]. However, evidence is accumulating that extensive crosstalk in signaling networks adds to the complexity of CSCs’ pathogenesis. Thus, there are still limited liver CSC-specific therapeutic strategies.

The MAEL gene, initially identified in Drosophila, plays an essential role in the miosis and construction of oocyte polarity [23]. In normal human tissues, MAEL is predominantly expressed in testis [24,25]. In contrast, MAEL has been found abnormally expressed in HCC [26], esophagus cancer [27], bladder urothelial carcinoma [28], and colorectal cancer [29]. Our previous study found that MAEL could upregulate the mRNA expression of stemness-related genes and CSC surface markers in HCC [26]. Consistent results have also been reported by other laboratories in esophagus and colorectal cancers [27,29]. Hence, we hypothesized that MAEL may be involved in maintaining HCC stemness properties. However, the molecular mechanism of MAEL regulation stemness function remains unclear.

## 2. Materials and Methods

### 2.1. Cell Lines and Cell Culture

Human HCC cell lines Huh7, HepG2, Hep3B, hepatoma cell line PLC/PRF/5 (8024), and the hepatocyte immortalized cell line MIHA were obtained in ATCC. The cell lines were verified by STR profiling and tested without Mycoplasma. The cells were cultured in DMEM high-glucose medium (Gibco, Thermo Fisher Scientific, Waltham, MA, USA), supplemented with 10% FBS (Gibco), 1% Penicillin (500 U/mL), and streptomycin (500 µg/mL). Lentiviral production cell line 293FT was purchased from Invitrogen, which grew with 10% FBS, 500 µg/mL geneticin (Invitrogen, Waltham, MA, USA), 1% non-essential amino acids (NEAA, Gibco), and 1 mM Sodium Pyruvate (Gibco) in DMEM medium. The cells were cultured in a 5% CO_2_ incubator at 37 °C.

### 2.2. Establishing Stable Knockdown and Knockout Cell Lines

For MAEL knockout, human MAEL (NM_032858.2)-targeting sgRNA/Cas 9 all-in-one plasmids (Genecopia, Rockville, MD, USA) were transfected into the cells. After 72 h, mCherry signal-sorted cells were seeded into 96-well plates for the generation of single clones. The knockout efficiency was examined by the IndelCheck kit (Genecopia). The MAEL depletion was evaluated by Western blot. For MAEL silencing, short hairpin RNA (shRNA) scramble or MAEL-targeting plasmids (BioLing Limited, Hong Kong SAR, China) and package plasmids were transfected into lentivirus in the 293FT cell line with ScreenfactA (InCella, Baden-Württemberg, Germany) to produce lentivirus particles. Stably transfected clones were selected with puromycin (Sigma-Aldrich, St. Louis, MO, USA).

### 2.3. Establishing Gene Overexpression Stable Cell Lines

Plasmids pLenti6-MAEL and pLenti6-PTGS2 were generated by TOPO-pLenti6 cloning kits (Invitrogen) according to the manufacturer’s instructions. Briefly, PLenti6-MAEL, pLenti6-PTGS2, pLenti6 vector plasmids, and package plasmids were transfected in the 293FT cell line to produce lentivirus particles. Stable transfection cells were infected with lentivirus and selected with blasticidin (Sigma-Aldrich).

### 2.4. RT-PCR and Q-PCR

Total RNA was extracted using TRIzol reagent (Takara, Kusatsu, Shiga, Japan) and reverse-transcripted into cDNA using a PrimeScript RT Master kit (Takara). Q-PCR was conducted using TB Green Premix Ex Taq II (Takara) and detected by ABI 7900HT (Applied Biosystems, Waltham, MA, USA). The primers are listed in Appendix A. The relative mRNA expression level was calculated by the 2^−ΔΔCt^ method using 18S RNA as an internal control.

### 2.5. Western Blot

Quantified protein lysates were resolved on SDS-PAGE (Bio-Rad, Hercules, CA, USA) and blotted onto 0.45 µm PVDF membranes (Merck Millipore, Burlington, MA, USA). The membranes were incubated with anti-human β-actin, CD133, CD44, MAEL, c-Myc, KLF4, IL-8, STAT3, p-STAT3, Akt, and p-Akt, followed by incubation with HRP secondary antibodies (Appendix A). The blots were exposed in X-ray film (Carestream, Rochester, NY, USA) with ECL (Bio-Rad).

### 2.6. Bioinformatic Analysis in TCGA Database

The LIHC transcriptome sequencing data and patients’ clinical information of The Cancer Genome Atlas (TCGA) were retrieved from Broad GDAC Firehose (Broad Institute, Cambridge, MA, USA) through the R program (version 3.5.3; R Foundation for Statistical Computing, Boston, MA, USA). The MAEL expression levels in 371 human HCCs and 50 non-tumor sample cases were analyzed. Meanwhile, according to the 50% cutoff value, the TCGA cohort was divided into high or low MAEL expression groups. The differences between the MAEL expression and clinicopathological features, as well as overall survival, were examined.

### 2.7. Cell Proliferation Assay

The cell proliferation ability was determined by XTT assay and foci formation. For the XTT assay, the cells were cultured at 1000 cells per well in 96-well plates. Absorbance was measured with the XTT assay kit (Sigma-Aldrich, St. Louis, MO, USA) using a spectrophotometer at OD 492 nm. For the foci formation, cells grown in 6-well plates for two weeks were fixed with 75% ethanol and stained with crystal violet (Sigma-Aldrich).

### 2.8. Migration and Invasion Assay

In vitro cell migration and invasion assays were performed in 8 µm pore size transwell chambers (BD Biosciences) and the same pore size Matrigel-coated chambers, respectively. According to the manufacturer’s instructions, the cells were placed into the inserted chambers with serum-free medium in 24-well plates, which were supplied with complete nutrition medium and incubated for 48~72 h depending on the cell type. The cells were fixed and stained with crystal violet before mounting.

### 2.9. Self-Renewal Assay

The self-renewal assays were conducted by colony formation and spheroid formation experiments. Cells (2~4 × 10^4^) were grown in 0.35% agar in 6-well plates pre-coated with 0.5% agar. The top of the agar was refreshed by medium every three days for 2~3 weeks. For the spheroid formation assay, the cells were seeded in low attachment plates (Corning, Corning, NY, USA) with special growth factor bFGF (Gibco), EGF (Gibco), B-27 (Gibco), and insulin (Gibco) in 2.5% methylcellulose (MC, Sigma-Aldrich) medium. Similarly, the spheroids were refreshed every three days and cultured at 37 °C for two weeks. The first passage of the spheroids was captured and then detached by TrypLE (Invitrogen) and passaged again with the same procedure. The formations of colonies and spheroids were captured by microscope (Olympus, Tokyo, Japan) under a digital camera.

### 2.10. Tumor Formation in Nude Mice and Sorafenib Treatment

Tumor cells were subcutaneously injected into the left and right dorsal flanks of mice individually. The tumor volumes were measured every week using the formula: L × W^2^ × 1/2. Treatment was initiated on day 5 post-tumor inoculation, and the mice were intragastrically injected with 20 mg/kg sorafenib every three days. The tumor volumes were monitored on days 5, 10, 20, and 30 for a total of 30 days. After being cultured, the mice were sacrificed, and the tumors were dissected for further analysis.

### 2.11. Flow Cytometry Analysis

The cells were labeled with PE-conjugated mouse anti-human CD133 (Miltenyi Biotec, North Rhine-Westphalia, Germany) and the respective isotype control to analyze the percentage of CD133^+^ cells. PI and FITC conjugated Annexin V (BD Biosciences, Franklin Lakes, NJ, USA) were used for early and late-stage apoptotic cell detection, according to the manufacturer’s instructions. The samples were analyzed on a BD FACSCanto II (BD Biosciences) or cell sorter FACS Aria SORP (BD Biosciences), and the data were evaluated using FlowJo 7.6 software (Tree Star Inc., Ashland, OR, USA).

### 2.12. RNA Sequencing and Pathway Analysis

The total RNA was isolated and processed by Hiseq platform (Navogene, Tianjin, China). The normalized gene expression level was calculated with fragments per kilobase of the exon model per million mapped reads (FPKM). Genes with significant differential expressions of WT to KO were analyzed by DESeq, generating a volcano graph with defined cutoff values that were adjusted for a *p* value of less than 0.01, and log2 (Fold Change) values of more than 2 or less than −2 were defined as upregulated or downregulated, respectively. The differential genes were further processed by the Kyoto Encyclopedia of Genes and Genomes (KEGG) pathway enrichment and Gene Set Enrichment Analysis (GSEA).

### 2.13. Immunofluorescent Staining

The cells were fixed and incubated with PE-conjugated CD133 or NFκB p65 primary antibodies, followed by incubation with Alexa Fluor 555-conjugated secondary antibodies for antibodies without fluorescent conjugation. The cells were counterstained with DAPI (Thermo Fisher Scientific) and visualized under a fluorescence microscope (Leica, Wetzlar, Germany).

### 2.14. Luciferase Reporter Assay

A fragment of 2500 bp (−2 kb–500 bp) PTGS2 promoter was cloned into the pGL3-basic vector (Promega, Madison, WI, USA). The pGL3-PTGS2, pLenti6 vector or pLenti6-MAEL and Renilla plasmid (5:5:1) were co-transfected into HEK293FT cells via lipofectamine 2000 (Invitrogen). The expression of the luciferase reporter gene was then measured with a Dual-Luciferase Reporter kit (Promega). To discover the binding sites of the PTGS2 promoter, a luciferase assay was conducted on a series of 5′ deletion PTGS2 promoter (T1-T4) pGL3 plasmids and MAEL-expressing constructs. To further confirm the binding sites, mutagenesis of the CEBPB binding site was constructed and detected with the luciferase activity.

### 2.15. IL-8 Detection with ELISA

An equal number of cells were seeded into a T25 flask with the same volume medium. An IL-8 concentration in the cell culture supernatant was determined by a Human IL-8 kit (Abcam, Cambridge, UK) according to the manufacturer’s instructions.

### 2.16. Statistical Analysis

All experiments were conducted three times. GraphPad Prism and SPSS software were applied for statistical analyses. Survival curves were generated by the Kaplan–Meier method and analyzed by the log-rank test. The significance level between each group was determined by Student’s *t*-test. A Chi-square test was used to compare the difference between MAEL expression and clinicopathological data. The RNA-Seq data were analyzed using R software (version 3.5.3; R Foundation for Statistical Computing, Boston, MA, USA) with the proper statistical packages. A statistical significance at the value of *p* < 0.05 was defined as a difference. The data are represented by the mean ± SD.

## 3. Results

### 3.1. HCC Patients with High-Level MAEL Display Poor Outcomes

Compared with non-tumor tissues, the expression of MAEL in tumor tissues was significantly elevated in the LIHC cohort and 50 paired HCC/adjacent non-tumor tissues (Figure 1A,B). The Kaplan–Meier analysis indicated that a high expression of MAEL was associated with poor overall survival compared with low-MAEL-expression HCC patients (Figure 1C). These observations are consistent with our previous findings that MAEL was upregulated in HCC in a separate cohort consisting of 91 pairs of tumor and adjacent non-tumor specimens [26]. Furthermore, it was determined that MAEL is associated with the male sex (Appendix A). The expression of MAEL was also detected in HCC cell lines and the immortalized liver cell line MIHA for subsequent functional assays (Figure 1D).

### 3.2. MAEL Promotes Aggressiveness in Liver Cells

To determine the pathological role of MAEL in HCC, we knocked out the MAEL gene in PLC8024 cells with the highest expression levels using CRISPR-Cas9 (Figure 2A and Appendix A). MAEL knockout significantly undermined stem-like features in the HCC cells, including slowing down the proliferation (Figure 2B,C) and attenuating the self-renewal ability of PLC8024 cells, as demonstrated by decreases in the number of colonies and spheroids (Figure 2D,E). Furthermore, the MAEL knockout cells showed declined mobility with fewer migrated (Figure 2F) and invaded cells (Figure 2G) in the transwell assay compared with wildtype cells. To access the role of MAEL in HCC formation, MAEL KO or WT cells were subcutaneously injected into nude mice. The deletion of MAEL led to a marked reduction in the tumor masses in mice (Figure 2H). Similar functional phenomena were achieved in PLC8024 cells with MAEL silenced by the lentivirus shRNA approach (Appendix A).

For completeness, we stably overexpressed MAEL in MIHA and Huh7 cells through lentivirus infection (Appendix A). The enhanced oncogenic phenomena were observed in the MAEL overexpression groups, including increased cell growth and foci formation, as well as the number of colonies and spheroid formation (Appendix A). The transwell assay indicated that MAEL expression drastically increased the invasion and motility ability in HCC (Appendix A). The overexpression of MAEL in MIHA and Huh7 cells resulted in larger tumor masses in nude mice (Appendix A). Collectively, these data support the oncogenic role of MAEL in HCC.

### 3.3. MAEL Promotes Stem-Cell-like Properties in HCC

MAEL regulated aggressive cancer features and the stem/progenitor cell characteristics of HCC. The knockout and overexpression of MAEL resulted in a respective concomitant decreased and increased expression of stemness-associated genes, including CD44, C-Myc, CD133, Oct-4, and Sox-2 in mRNA, and protein levels (Figure 3A,B), suggesting that MAEL may regulate the stemness of HCC. We accessed the liver CSC marker CD133 on MAEL-manipulated cells by FACS and immune fluorescent staining. An enrichment of CD133^+^ subpopulation cells was observed in MAEL-transfected cells, while reduced CD133^+^ cells were determined in MAEL knockout cells (Figure 3C,D).

### 3.4. MAEL Contributes to the Formation of Resistance to Sorafenib in HCC Cells

Sorafenib is one of the first-line treatments for advanced hepatocellular carcinoma, yet its survival benefit is only about three months [3]. This unsatisfactory efficacy is partially due to the drug resistance connected to the cancer stemness properties in HCC [30]. Herein, we investigated the contribution of MAEL on sorafenib resistance in HCC. Upon sorafenib treatment, decreased cell viability and IC_50_ were observed in the MAEL knockout of PLC8024 cells, while opposite results were achieved in MAEL overexpression cells (Figure 4A). Annexin V/PI analysis showed that MAEL deletion augmented cell apoptosis in PLC8024 cells treated with sorafenib, while the reverse phenotype was observed in Huh7 MAEL-expression cells (Figure 4B).

We further explored the therapeutic potential for targeting MAEL alone and its combined effect with sorafenib in vivo. Nude mice subcutaneously inoculated with MAEL-silenced PLC8024 cells were administered intragastrically with the mouse model’s relevant dose of sorafenib (Figure 4C). Consistently, we observed a remarkable decrease in tumor volume in the mice implanted with MAEL-silencing cells or treated with sorafenib alone. Strikingly, an enhanced antitumor effect was achieved in mice bearing MAEL-silencing cells in combination with sorafenib treatment. Mice that received the combination treatment showed more pronounced tumor suppression compared with the control group, and tumor masses even disappeared in two of the five mice in the combination group (Figure 4D). Taken together, these data indicate a potential therapeutic strategy for HCC patients by blocking MAEL in combination with sorafenib.

### 3.5. MAEL May Regulate HCC Stemness and Sorafenib Resistance through IL-8/STAT3/NF kB/AKT Signaling

To understand the mechanism by which MAEL regulates HCC stemness and sorafenib resistance, we conducted RNA sequencing on MAEL-manipulated cell lines. KEGG analysis indicated that MAEL knockout deregulated genes were closely associated with stem cell pluripotency regulation signaling, cytokine–cytokine receptor interactions, and PI3K-Akt pathways (Figure 5A). In addition, the gene set enrichment assay (GSEA) showed that deregulated genes by MAEL knockout were enriched in the Wong embryonic stem cell core (Figure 5B) [31]. Among the dysregulated genes, the CSC-related gene PTGS2 was listed in the top five downregulated genes by MAEL knockout (Figure 5C). Prostaglandin-endoperoxide synthase 2 (PTGS2), also known as cyclooxygenase-2 (COX-2), is characterized as a rate-limiting enzyme catalyzing the conversion of arachidonic acid to prostaglandins [32]. PTGS2 has been implicated in tumorigenesis and recognized as a potential target for the management of cancer [33,34]. As evidenced by qPCR, PTGS2 was downregulated in MAEL knockout PLC8024 cells, whereas opposite expression patterns were found in the overexpression cells (Figure 5D). Since MAEL has been implied to have a nuclear function in regulating gene expression [23], we next investigated whether MAEL may transactivate the expression of PTGS2 with a luciferase reporting assay (Figure 5E). Luciferase activity in 293FT cells transfected a segment of PTGS2 promoter constructs, and MAEL plasmid was elevated in comparison with cells transfected with reporter and control vectors (Figure 5F), suggesting that MAEL drives PTGS2 transcriptional activity.

To further identify the binding site of the PTGS2 promoter region, luciferase activity on a series of 5′ deletion PTGS2 promoter (T1-T4) plasmids was conducted. We found that deletion from full-length to T2 (−2000/−933) had no effect on the PTGS2 promoter activity induced by MAEL, and further deletion from T2 to T3 (−933/−268) significantly suppressed the PTGS2 promoter activity, indicating that regions from T2 to T3 on the PTGS2 promoter are critical for the transactivation of PTGS2 in response to MAEL (Appendix A). Then, we conducted a prediction analysis of the cis-regulatory elements between the −933 and −268 regions of the PTGS2 promoter using JASPER [35], TRRUST [36], and PROMO [37], and identified one CEBPB binding site. The site-directed mutagenesis to the CEBPB binding site impeded the activation of the PTGS2 promoter by MAEL (Appendix A). Collectively, we demonstrated that the binding of CEBPB to the PGST2 promoter was required for the activated expression of PTGS2 mediated by MAEL.

To demonstrate whether MAEL drives tumor initiation and self-renewal by activating PTGS2 gene expression, PTGS2 was overexpressed in MAEL-KO 8024 cells to determine if the effect of MAEL deletion could be resecured upon PTGS2 overexpression. Using the lentiviral approach, PTGS2 was overexpressed in MAEL-KO 8024 cells, and the efficient PTGS2 overexpression was confirmed by Western blot analysis (Figure 6A). Elevated proliferative capability, foci formation, and colony and spheroid formation were observed in PTGS2-expressing MAEL-KO cells (Figure 6B–E), indicating that the ectopic expression of PTGS2 could restore HCC cell aggressiveness impaired by MAEL knockout.

Previous studies have indicated that MAEL promoter tumorigenesis is dependent on the AKT/NF-κb/IL8 signaling, which has been associated with CSC regulation [26,27,38]. Thus, we further hypothesized that MAEL-regulating HCC stemness is dependent on IL-8. As evidenced by Western blot, MAEL knockout suppressed the expression of IL-8 and the phosphorylation of Akt and STAT3, whereas it induced expression and activation in MAEL overexpression cells (Figure 7A). The decreased secretion of IL-8 in condition medium was confirmed in MAEL knockout cells (Figure 7B). The activation of NFκb through canonical p65 phosphorylation triggers the translocation of the NF-κb complex from cytoplasm to nuclei and transcriptionally activates downstream target genes, including IL-8 [39,40]. The nuclear translocation of NFκb (P65) was observed in Huh7 cells expressing MAEL (Figure 7C). Moreover, MAEL knockdown obviously decreased PTGS2 expression and AKT phosphorylation upon sorafenib treatment, which supports the critical role of MAEL in the transcriptional regulation of stemness-related genes against sorafenib working (Figure 7D). In summary, our results suggest that MAEL plays a critical role in driving HCC stemness via PTGS2/Akt/NF-κb/IL8 signaling and subsequently promotes self-renewal and drug-resistant phenotypes.

## 4. Discussion

Previous studies have identified MAEL as an oncogene promoting tumor aggressiveness in liver and provided clues indicating that MAEL could enhance self-renewal and chemoresistance capabilities through regulatory stemness-related genes [26]. However, the stemness-regulating mechanisms and the downstream targets of MAEL have not yet been documented. Thus, our present study further investigated the unique role of MAEL in regulating HCC stemness and revealed underlying molecular mechanisms that may facilitate the exploitation of clinical therapeutic strategies for targeting liver CSCs.

MAEL was restrictively detected in HCC tumors while hardly detected in non-tumor liver tissues in the TCGA cohort. The overexpression of MAEL was positively correlated with poor survival, which supports MAEL as a prognostic factor in HCC prognosis. Interestingly, elevated MAEL expression was observed dominantly in male HCC patients (124/184 cases in males, 61/187 in females, *p* < 0.001). Considering that HCC is strongly predominant in males, with an incidence 2–3 times higher in males than in females [32], whether there is a distinct role of MAEL in male HCC remains to be elucidated.

Through functional studies on modulated MAEL-expression cell lines, we found that MAEL could not only promote the ability of proliferation and mobility, but it also enhanced stemness activities, including elevating the CD133^+^ subpopulation and self-renewal HCC cells. As we previously reported, the CD133^+^ HCC subpopulation has been identified as liver CSCs with enhanced resistance to therapeutics [41]. We also discovered that MAEL supports the development of resistance to sorafenib treatment both in vitro and in vivo. Collectively, our data strongly support that MAEL plays a pivotal role in maintaining liver CSC features. Evidence is accumulating that RNAi therapeutics delivered by nanoparticles hold great potential in oncology [42]. Considering the potential therapeutic efficacies of MAEL knockdown in our findings, MAEL siRNA nanoparticle therapeutics with sorafenib or other molecular-targeted agents may enhance the options available for treating HCC in the future.

Though MAEL-promoting tumorigenicity and metastasis are closely correlated with PI3K-AKT signaling, the molecular mechanisms of MAEL in regulating cancer stemness have remained unclear thus far. In ESCC, MAEL is closely associated with PI3K-AKT signaling and NFκB (p65) activation, which results in IL-8 secretion in ESCC [27]. IL-8 has been involved in the maintenance of cancer stemness properties through the activation of STAT3 [43,44]. Considering that promoters of stemness-associated genes Klf4, Sox2, and C-Myc have been found occupied by transcription factor STAT3 for gene activation [45,46], it is conceivable that IL-8/STAT3 may be involved in MAEL-dependent stemness regulation. Indeed, we observed that MAEL induced the activation of AKT and STAT3 through phosphorylation and enhanced IL8 secretion. The effect of MAEL on the activation of AKT/STAT3/IL8 signaling was further confirmed by the translocation and activation of NFκb. In summary, we identified the contribution of the AKT/NFκb/STAT3/IL8 pathways to MAEL-mediated stemness in HCC.

Of note, we identified PTGS2 as a downstream target of MAEL and a key mediator of MAEL-dependent cancer stemness through AKT activation. PTGS2 was implicated in the regulation of CSCs in bladder and urothelial cancers [47]. Recently, it was found that PTGS2 regulates HIF2α under hypoxic conditions to promote tumor development and resistance to sorafenib in HCC [48]. In fact, there are several PTGS2-specific inhibitors available in clinical treatment, including FDA-approved Celebrex and Bextra. Considering the role of MAEL in CSC regulation and the substantial effect of MAEL-targeting in impairing HCC tumor growth in mice, alone or in combination with sorafenib, it raises the possibility of the PTGS2 inhibitor as a combination treatment along with already available therapies for anti-cancer stemness-targeting in HCC [49,50]. However, whether the PTGS2 inhibitor as a combination treatment with already available therapies would bring clinical benefits for HCC patients needs further investigation.

There may be several limitations in this work. The human MAEL protein contains a high-mobility-group (HMG) domain in its N-terminal segment that is known to mediate DNA binding. Indeed, we identified PTGS2 as a downstream target transactivated by MAEL through the binding of CEBPB to the PTGS2 promoter. However, due to the time limitation, we did not confirm the DNA binding ability and specificity of MAEL to its downstream target genes. ChIP-PCR and electrophoretic mobility shift assay may be useful to fill in these gaps in the future.

## 5. Conclusions

In summary, we defined a novel function of MAEL in promoting liver CSCs and sorafenib resistance in HCC cells through the regulation of the PTGS2/AKT/STAT3 signaling cascade. Targeting MAEL-dependent cancer stemness through PTGS2 inhibitors could be a promising therapeutic strategy for HCC management and the reversal of sorafenib resistance in particular.

## Figures and Tables

**Figure 1 cancers-14-02880-f001:**
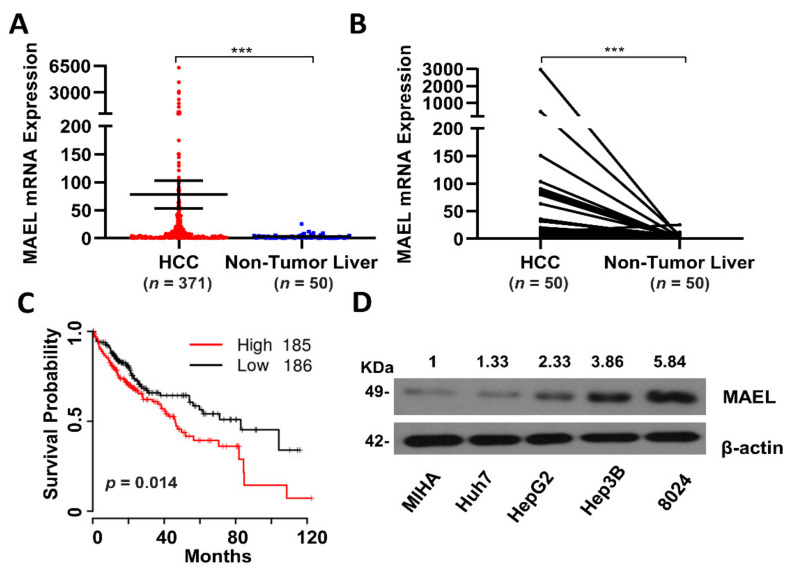
MAEL expression in HCC samples and cell lines. (**A**) The mRNA expression of MAEL analyzed from 371 HCC tissues and 50 non-tumor tissues in the TCGA database. (**B**) The mRNA expression of MAEL in 50 HCC tissues and matched adjacent non-tumor tissues analyzed in the TCGA database. (**C**) Kaplan–Meier plot for overall survival of high (*n* = 185) and low MAEL (*n* = 186) expression groups (50% cut off) in the TCGA database. *p* = 0.014. (**D**) Western blots of MAEL expression in HCC cell lines. Full Western Blot can be found in Appendix A. *** *p* < 0.001.

**Figure 2 cancers-14-02880-f002:**
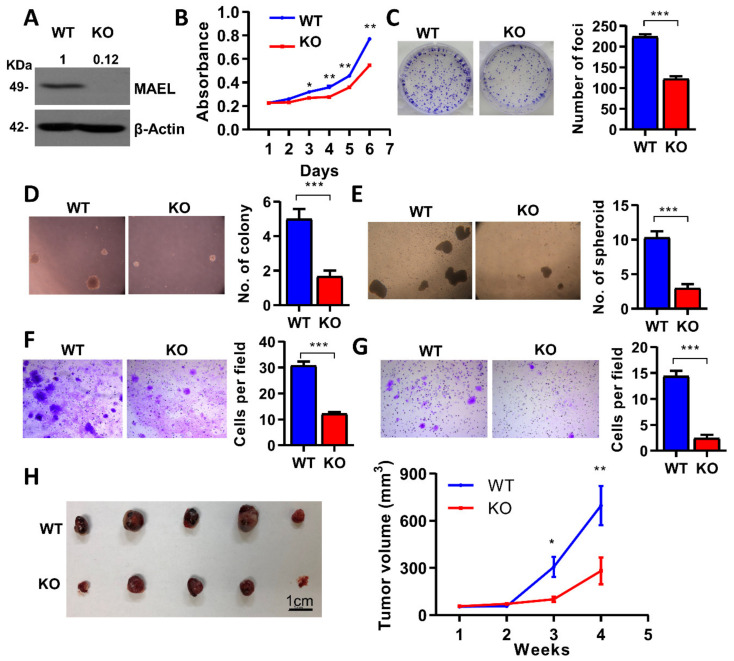
Deletion of MAEL-impaired aggressive phenotype in HCC. (**A**) Western blot validation of MAEL knockout in PLC8024 cells. β-Actin served as an internal control. Full Western Blot can be found in Appendix A. (**B**) Cell proliferation rate determined by XTT assay. Representative images and quantitation of (**C**) foci formation, (**D**) colony formation, and (**E**) spheroids in soft agar in MAEL wildtype and knockout PLC8024 cells. Representative images and quantitation of (**F**) migrated and (**G**) invaded cells in MAEL wildtype and knockout PLC8024 cells. (**H**) Representative images and tumor volumes of xenograft tumors derived from mice subcutaneously inoculated with wildtype or knockout cells (*n* = 5 per group). WT, wildtype; KO: MAEL knockout. Scale bar stands for 1 cm. The values indicate the mean ± SD of three independent experiments. * *p* < 0.05; ** *p* < 0.01; *** *p* < 0.001.

**Figure 3 cancers-14-02880-f003:**
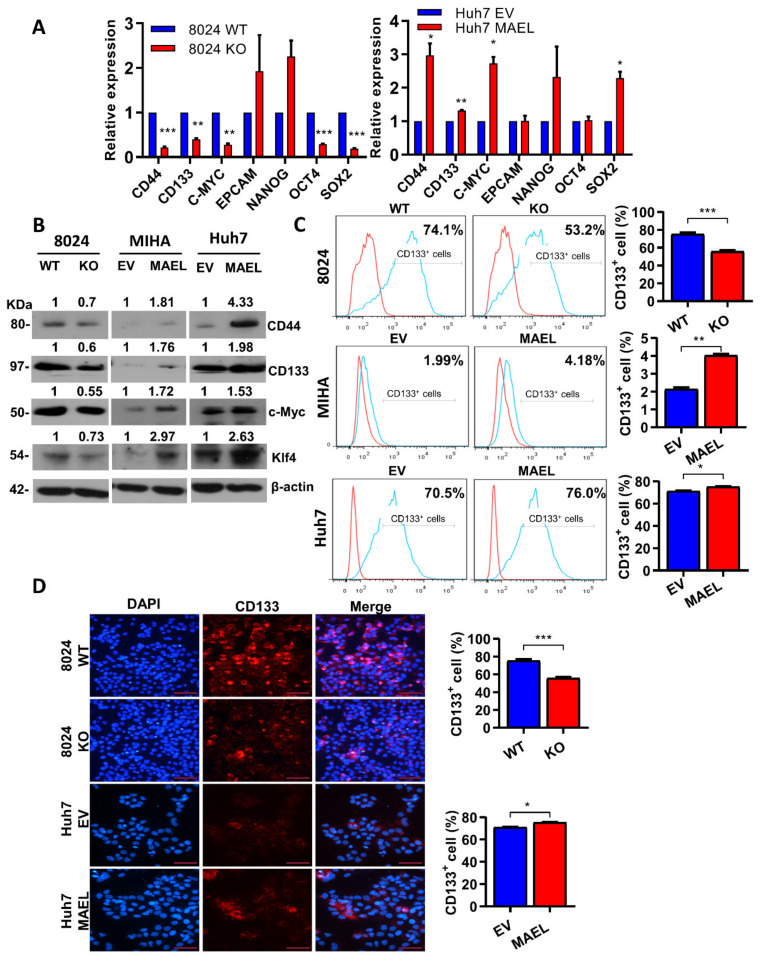
MAEL increased stemness-associated gene expression in HCC cells. Relative mRNA (**A**) and protein (**B**) expressions of stemness-related genes in MAEL overexpression or knockout cells. β-Actin and 18s RNA served as a loading control. Full Western Blot can be found in Appendix A. (**C**) Representative flow cytometry histogram and quantification of CD133^+^ population in cells with or without MAEL modulation. (**D**) Representative immunofluorescent images and quantification of number of CD133^+^ PLC8024 and Huh7 cells with or without MAEL modulation. Scale bar stands for 50 µm. The values indicate the mean ± SD of three independent experiments. * *p* < 0.05; ** *p* < 0.01; *** *p* < 0.001.

**Figure 4 cancers-14-02880-f004:**
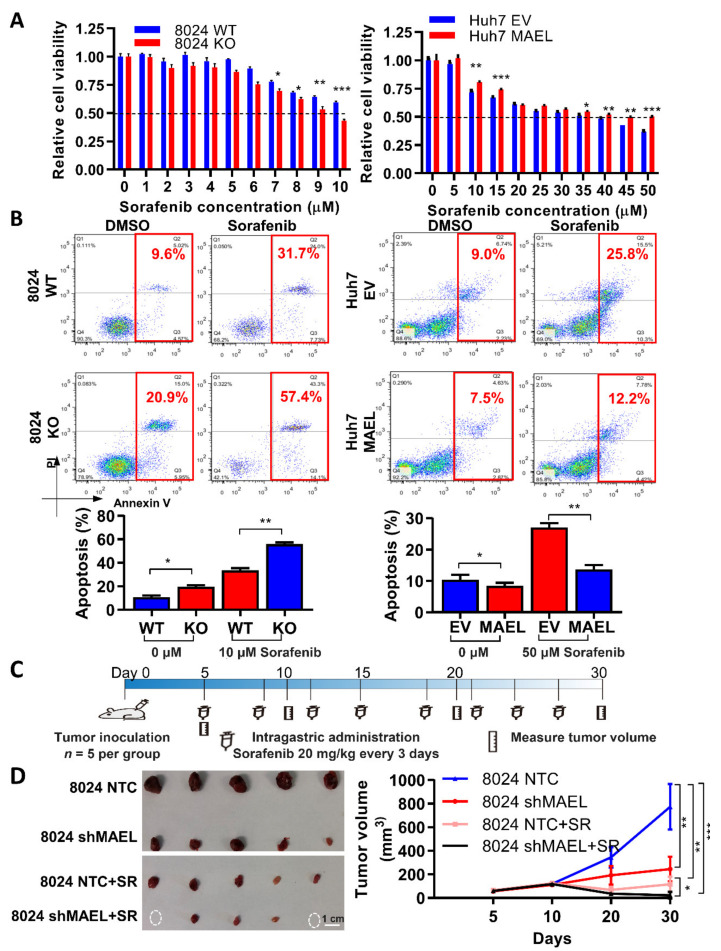
MAEL regulates HCC cell sensitivity to sorafenib. (**A**) XTT assay indicates cell viability upon sorafenib treatment in cells with MAEL overexpression or knockout. WT, nontarget control; KO, MEAL gene knockout; EV, empty vector control. (**B**) Representative Annexin V/PI staining dot plots and quantification of apoptosis in cells with or without sorafenib treatment. (**C**) Schematic illustration of sorafenib treatment in mice implanted with PLC8024 scramble and MAEL shRNA cells. (**D**) Representative images and tumor volume formed in mice subcutaneously injected with MAEL modulation cells with or without sorafenib treatment (*n* = 5 per group). The circles stand for no tumor formatted. Scale bar stands for 1 cm. Data represent the mean ± SD of three independent experiments. * *p* < 0.05; ** *p* < 0.01; *** *p* < 0.001.

**Figure 5 cancers-14-02880-f005:**
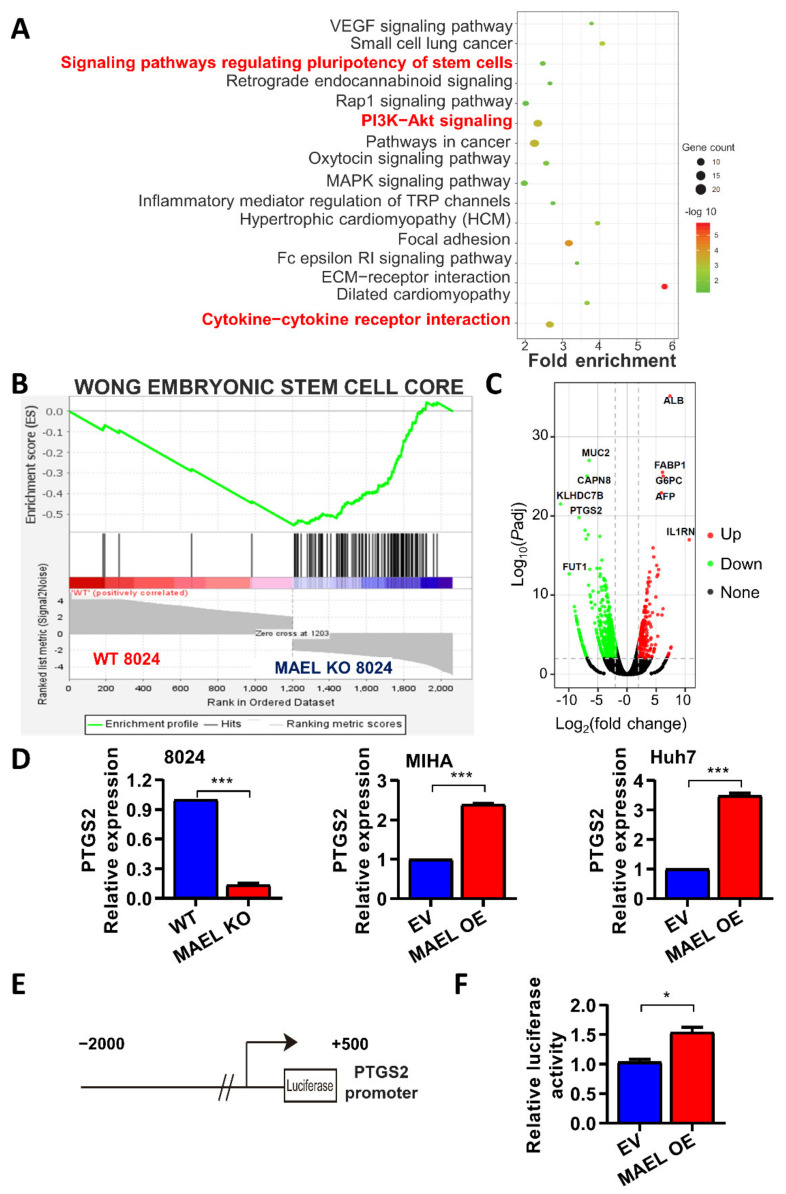
PTGS2 is transcriptionally activated by MAEL and involved in the MAEL-mediated regulation of stemness in HCC. (**A**) The plot of KEGG pathways in MAEL wildtype versus knockout PLC8024 cells. The color and size of the bubble represent the enrichment significance and gene numbers enriched in a pathway, respectively. (**B**) Gene set enrichment analysis (GSEA) demonstrates MAEL modulation significantly enriched in embryonic stem cell core. (**C**) The volcano diagram illustrates differentially expressed genes by MAEL knockout. Spots indicate genes differentially expressed between the two groups (red, upregulated; green, downregulated; gray, not significant genes; –Log_10_Padj > 2, |Log_2_Fold change| > 2). (**D**) Relative mRNA level of PTGS2 validated by qRT-PCR in MIHA, 8024, and Huh7 cells. (**E**) Schematic illustration of the luciferase reporter constructs containing PTGS2 promoter. (**F**) Relative luciferase activity in 293FT cells co-transfected with PTGS2-luciferase reporter and MAEL. Data represent the mean ± SD of three independent experiments. * *p* < 0.05; *** *p* < 0.001.

**Figure 6 cancers-14-02880-f006:**
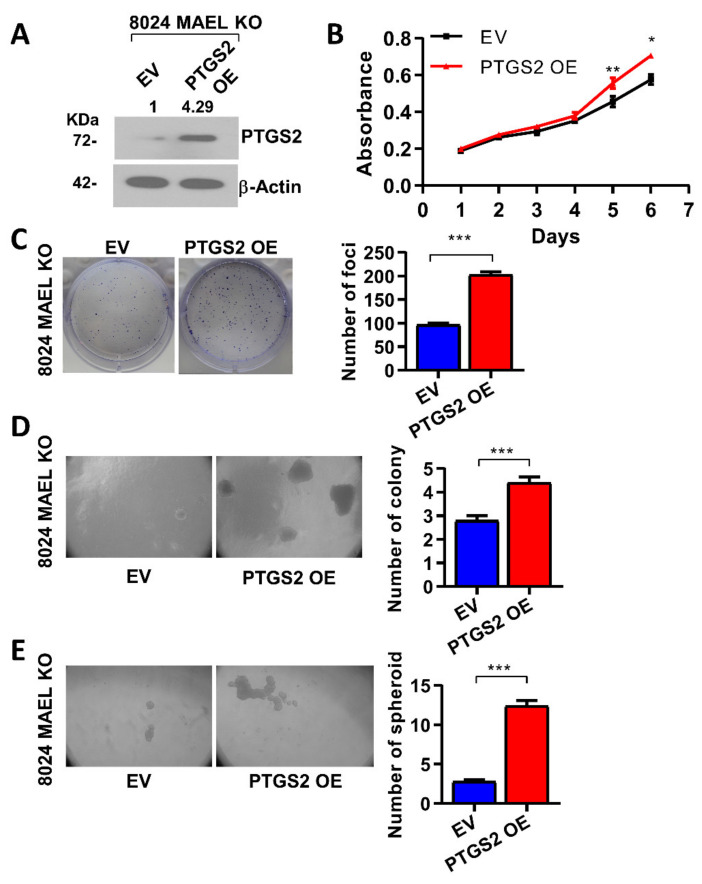
PTGS2 restores cancer properties in MAEL-deletion cells. (**A**) Western blots showed ectopic expression of PTGS2 in MAEL knockout 8024 cells. β-Actin served as internal control. Full Western Blot can be found in Appendix A. (**B**) Growth curve of PTGS2 overexpression cells determined by XTT assay. Representative images and quantitation of (**C**) foci formation, colony formation in soft agar (**D**), and (**E**) spheroid formation in MAEL knockout 8024 cells with or without PTGS2 overexpression. The values indicate the mean ± SD of three independent experiments. * *p* < 0.05; ** *p* < 0.01; *** *p* < 0.001.

**Figure 7 cancers-14-02880-f007:**
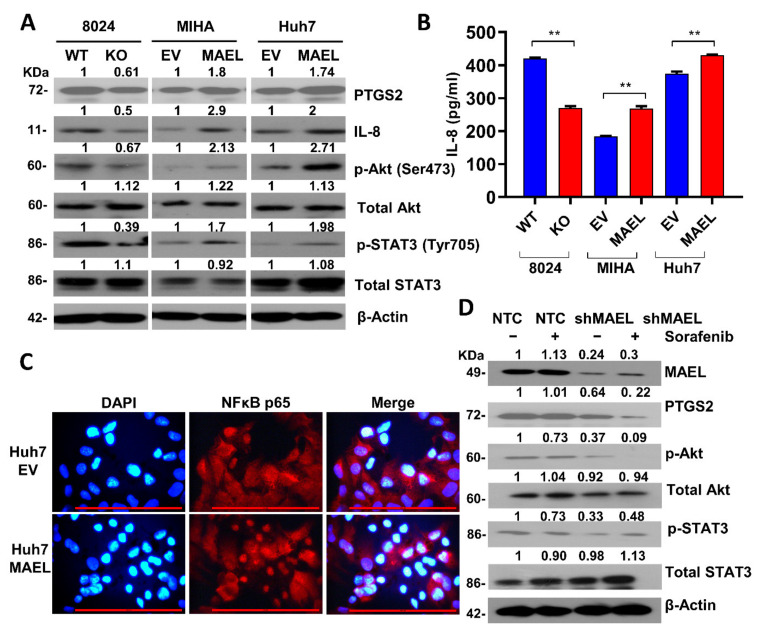
MAEL activates the IL8/Akt//NFκB/STAT3 signaling pathway. (**A**) Immunoblots of PTGS2, IL-8, STAT3, p-STAT3, AKT, and p-Akt in cells with or without modulated MAEL expression. Full Western Blot can be found in Appendix A. (**B**) Quantitative analysis of IL-8 concentration determined by ELISA in MAEL expression-modulated cells. (**C**) Representative images of immunofluorescence of Huh7 cells transfected with MAEL or vector cells. Cells were stained with NF-κB (P65) antibody (red). Nuclei were labeled by DAPI (blue). Scale bar stands for 50 µm. (**D**) Immunoblots of pAkt, pSTAT3, and PTGS2 in PLC8024 cells treated with sorafenib with or without MAEL silencing. Data represent the mean ± SD of three independent experiments. ** *p* < 0.01. Full Western Blot can be found in Appendix A.

## Data Availability

RNA-seq data were uploaded to the GEO database (GSE175454). Other supporting data were uploaded to the HKU Data Repository. https://doi.org/10.25442/hku.14497815.v1 (accessed on 6 May 2021).

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
