# Peer review of "MAEL Augments Cancer Stemness Properties and Resistance to Sorafenib in Hepatocellular Carcinoma through the PTGS2/AKT/STAT3 Axis"

_cancers, 2022, doi:10.3390/cancers14122880_

Round 1

Reviewer 1 Report

Interesting study on an important topic in hepatocellular carcinoma. At the same time, we believe some points should be better discussed in the revised manuscript. 

The manuscript is quite well written and organized. English could be improved.

The figures and tables are comprehensive and clear.

The introduction explains in a clear and coherent manner the background of this study.

We suggest the following modifications:

  • Introduction section: although the authors correctly included important papers in this setting, we believe some studies regarding novel and emerging treatments in hepatocellular carcinoma should be cited within the introduction, only for a matter of consistency. We think it might be useful to introduce the topic of this interesting study.
  • Methods and Statistical Analysis: nothing to add.
  • Discussion section: Very interesting and timely discussion. Of note, the authors should expand the Discussion section, including a more personal perspective to reflect on. For example, they could answer the following questions – in order to facilitate the understanding of this complex topic for readers: what potential does this study hold? What are the knowledge gaps and how do researchers tackle them? How do you see this area unfolding in the next 5 years? We think it would be extremely interesting for the readers.

However, we think the authors should be acknowledged for their work. In fact, they correctly addressed an important topic in HCC, the methods sound good and their discussion is well balanced.

One additional little flaw: the authors could better explain the limitations of their work, in the last part of the Discussion.

We believe this article is suitable for publication in the journal although some revisions are needed. The main strengths of this paper are that it addresses an interesting and very timely question and provides a clear answer, with some limitations.

We suggest a linguistic revision and the addition of some references for a matter of consistency. Moreover, the authors should better clarify some points.

Reviewer 2 Report

Comment on the manuscript cancers-1699648 by Shi, et al.

MAEL plays a central role during spermatogenesis by repressing transposable elements and preventing their mobilization, while this molecule is highly expressed in several types of cancer. The authors investigated the association between the MAEL expression and patients’ survivals. In addition, they showed that PTGS2 is a downstream target of MAEL, which resulted in IL-8 secretion and activating AKT/NF-kB/STAT3 pathway. The experiments were well designed and provided detailed biochemical mechanism on the role of MAEL in PTGS2/AKT/STAT3 signaling.

There are several questions and suggestions for corrections:
1) Line 69‒78 in the last paragraph of Introduction is a summary of this study, which is not adequate for Introduction.

2) The cut-off value of MAEL is not adequate (the 50% high or low expression groups). It should be determined using ROC curve analysis.
3) Supplementary Figure 3A‒H (line 239‒247) should be described as formal Figure in the text.
4) In Figure 5B, gene expression of MAEL knockout in 8024 cells were enriched in Wong embryonic stem cell core by GSEA. Then those of MAEL enhancement (MAEL OE in MIHA and Huh7 cells) should be analyzed. Also volcano diagram of gene expression of these cells should be shown.

5) Luciferase assay in Figure 5E is crude. At least, binding sites of MAEL in the PTGS2 promoter should be discussed.

6) PTGS2 and COX2 should be unified.

7) The alignment of the molecules in the WB (Figure 7A and D) should be modified in accordance with Graphical Abstract.

8) The first sentence of Discussion should be in Introduction, which could be removed.

9) Are there any reports which demosntrated that sorafenib targeted COA2/AKT/STAT3 kinase?

Reviewer 3 Report

In this study, Shi et al. investigated a role for Maelstrom (MAEL) in hepatocellular cancer (HCC) stemness and resistance to chemotherapy, specifically sorafenib. The authors used HCC (PLC8024 and Huh7) and normal liver (MIHA) cell lines to explore the effect of MAEL silencing or overexpression on cell proliferation, cancer stem cell (CSC) gene expression and function, and sensitivity to sorafenib in culture and in xenograft tumor mouse models. They identified PTGS2 (COX2) as an important downstream target of MAEL, as well as activation of IL-8 secretion and AKT/NF-kB/STAT3 signaling pathway.

Overall, the manuscript is well written, and the results are interesting, well presented, and thorough.

Specific comments: 

  1. “Kaplan-Meier analysis indicated that high expression of MAEL is associated with poor overall survival of HCC patients” (lines 202-203). Both outcomes could be considered poor, so it would be better to describe this result as high MAEL expression is associated with worse overall survival compared to low MAEL expression.
  2. There is a technical issue with the Western blot data presented in Figure 1D, in particular with the upper panel (MAEL) in the first lane (MIHA cells). A “half band” like this is usually due to a transfer/botting issue or the lane actually being cut. From the images of the original blots, it looks like outer lanes on both sides are problematic. This Western blot needs to be redone and the figure replaced.
  3. “a synergistic inhibition effect was achieved in mice baring MAEL silencing cells in combination with sorafenib treatment” (lines 279-280). While clearly the tumors in the combination group (shMAEL + sorafenib) are the smallest, this appears to be an additive effect rather than a “synergistic” interaction. The author should edit this sentence accordingly.

Round 2

Reviewer 1 Report

The authors addressed most of the queries we raised.

We recommend the inclusion of the following studies, only for a matter of consistency (PMID: 32402160 ; PMID: 34429006)

Author Response

Thanks for your great suggestion on improving the accessibility of our manuscript. We inserted ten new references in the revised manuscript highlighted in yellow. PMID: 32402160 was cited as refernce10. We add citations in novel monotherapy and immunotherapy drugs for hepatocellular carcinoma in the Introduction section. Also, references for the transcription factor prediction websites were cited. We carefully checked that all the cited references were relevant to our research.

Once again, thank you very much for your comments and suggestions.